# Reproductive Consequences of Electrolyte Disturbances in Domestic Animals

**DOI:** 10.3390/biology11071006

**Published:** 2022-07-03

**Authors:** Elżbieta Gałęska, Marcjanna Wrzecińska, Alicja Kowalczyk, Jose P. Araujo

**Affiliations:** 1Department of Environmental Hygiene and Animal Welfare, Wrocław University of Environmental and Life Sciences, Chełmońskiego 38C, 51-630 Wrocław, Poland; 113226@student.upwr.edu.pl (E.G.); marcjanna.wrzecinska@zut.edu.pl (M.W.); 2Mountain Research Centre (CIMO), Instituto Politécnico de Viana do Castelo, Rua D. Mendo Afonso, 147, Refóios do Lima, 4990-706 Ponte de Lima, Portugal; pedropi@esa.ipvc.pt

**Keywords:** electrolytes, balance, reproduction, animals

## Abstract

**Simple Summary:**

Electrolyte balance is very significant, ensures homeostasis of the organism, maintains reactions, nerve conduction, and proper polarity of cell membranes, and electrolytes are important in fertility. It is known that disturbances in the electrolyte balance lead to reproductive dysfunction. In men, there is a decrease in sperm motility, loss of sperm capacitation, and male infertility. In turn, in women, changes in the composition of the follicular fluid are observed, leading to a restriction of follicular growth. Imbalance of oocyte electrolytes, resulting in a lack of oocyte activation and, consequently, infertility.

**Abstract:**

Electrolyte balance is essential to maintain homeostasis in the body. The most crucial electrolytes are sodium (Na^+^), potassium (K^+^), magnesium (Mg^2+^), chloride (Cl^−^), and calcium (Ca^2+^). These ions maintain the volume of body fluids, and blood pressure, participate in muscle contractions, and nerve conduction, and are important in enzymatic reactions. The balance is mainly ensured by the kidneys, which are an important organ that regulates the volume and composition of urine, together with which excess electrolytes are excreted. They are also important in the reproductive system, where they play a key role. In the male reproductive system, electrolytes are important in acrosomal reaction and sperm motility. Sodium, calcium, magnesium, and chloride are related to sperm capacitation. Moreover, Mg^2+^, Ca^2+^, and Na^+^ play a key role in spermatogenesis and the maintenance of morphologically normal spermatozoa. Infertility problems are becoming more common. It is known that disturbances in the electrolyte balance lead to reproductive dysfunction. In men, there is a decrease in sperm motility, loss of sperm capacitation, and male infertility. In the female reproductive system, sodium is associated with estrogen synthesis. In the contraction and relaxation of the uterus, there is sodium, potassium, and calcium. Calcium is associated with oocyte activation. In turn, in women, changes in the composition of the follicular fluid are observed, leading to a restriction of follicular growth. Imbalance of oocyte electrolytes, resulting in a lack of oocyte activation and, consequently, infertility.

## 1. Electrolyte Balance

Electrolytes are charged ions that are essential to live organisms, such as sodium (Na^+^), potassium (K^+^), calcium (Ca^2+^), and chloride (Cl^−^). The most important electrolytes are described in Table 1. They play a key role in maintaining electrical neutrality in cells, generating and conducting action potentials in nerves and muscles, and are important in enzymatic reactions [1,2]. Moreover, electrolytes are crucial for osmoregulation purposes in living organisms, thus electrolyte balance in the body of an organism is necessary for the normal function of cells and organs [2].

Electrolyte balance is related to the water balance of the organism [6]. Homeostasis between water and electrolytes is needed to coordinate neural pathways, including the regulatory center in the brain responsible for thirst and thermoregulation in the hypothalamus.

Fluid deficiency in the body increases the concentration of ions in extracellular fluids, increasing osmolarity and reducing plasma volume, and stimulating the center in the brain responsible for thirst, and another for kidney function [7]. Water deficiency slows the renal glomerular filtration rate and then the response of the renin–angiotensin–aldosterone system (RAAS), resulting in a reduction in sodium excretion and an increase in vasopressin release. The water intake restores the water balance. On the other hand, consuming too much fluid leads to a reduction in osmolarity and an increase in plasma volume [6,8,9]. Total body water (TBW) is divided into extracellular fluid (ECF) and intracellular fluid (ICF), of which ECF consists of three main compartments: plasma, connective tissue, and interstitial water, and ICF are the largest component [6,7,10]. Both electrolytes and water balance contribute to the physiological balance of water and mineral salt exchange processes between the body and the external environment and lead to the maintenance of control over the physiological processes and physicochemical mechanisms, i.e., to maintain homeostasis of the whole organism. Controlling the proper balance of water and electrolytes is crucial for the maintenance of the functions of internal organs, the production of the necessary cellular energy, and the occurrence of metabolic processes [11]. The reference values of selected electrolytes in the blood serum are presented in the Table 2.

The Table 3 presents the daily demand for selected electrolytes for animal species.

The kidneys play an important role in regulating the balance of water and electrolytes by regulating urine volume and the extraction of electrolytes [20,21]. Furthermore, these regulating processes take place under the influence of antidiuretic hormone (ADH), aldosterone (ALD), and other hormones by selective re-absorbing and secreting water and electrolytes in renal tubules, causing significant differences in the composition and concentration of urine and plasma [21]. This regulation aims to maintain the correct plasma osmolarity, electrolyte, and acid-base balance [21].

### 1.1. Anti-Diuretic Hormone (ADH)

Antidiuretic hormone (ADH) is synthesized in the hypothalamus and regulates water reabsorption and thirst sensation [10,22]. ADH is also known as vasopressin (AVP) [23] and plays a key role in maintaining the homeostasis of body fluids in the body by maintaining plasma osmolality at an appropriate level and controlling renal excess water excretion [8,23]. Table 4 presents the normal plasma osmolality parameters for selected animal species.

The main stimulus for the synthesis of vasopressin is dehydration, which increases plasma osmolarity [22,23]. Any changes in extracellular fluid volume are associated with changes in osmolarity and therefore ADH secretion [22]. There is also non-osmotic activation of vasopressin synthesis that occurs during physical activity or nicotine consumption [22]. The target site of vasopressin action is the renal collecting duct, which is rich in vasopressin V2R receptors and vascular smooth muscle cells [23]. Antidiuretic hormone acts on the kidneys by increasing cortex permeability of the cells of cortex and the core of the renal collecting ducts. This hormone binds to the V2 receptor, which is activated by cyclic AMP (cAMP) activated by adenylate cyclase. On the other hand, an increase in cAMP concentration activates phosphokinase A, which phosphorylates the aquaporin 2 water channels, which results in the reabsorption of water from the tubular lumen [22]. In pathological conditions such as diabetes insipidus, there is a lack of antidiuretic hormone in the body [23]. The kidneys are responsible for the re-absorption of water and sodium in the proximal tube. Water resorption is by diffusion and is dependent on sodium resorption and the action of vasopressin [7,28]. When the concentration of vasopressin in plasma is high, the permeability of water to the kidney increases. Vasopressin is influenced by cardiovascular receptors that are involved in AVP release. With high osmolarity, vasopressin secretion is stimulated, whereas low osmolarity inhibits AVP secretion. The RAAS system plays an important role in regulating the management of the body water [6,29].

### 1.2. The Renin–Angiotensin–Aldosterone System (RAAS)

The system responsible for the regulation of body water and blood pressure is the renin–angiotensin–aldosterone system [30,31]. Renin is an enzyme secreted by kidney juxtaglomerular cells that supply blood to the glomeruli and is regulated by pressure reduction in the afferent arterioles, neural stimulation, and feedback in the distal tubules [32]. Renin is involved in the conversion of angiotensinogen (synthesis in the liver) into angiotensin I, which is converted by the angiotensin-converting enzyme (ACE) (secreted in the lungs) into angiotensin II, which has a strong vasoconstrictive effect [29,33]. Next, angiotensin II stimulates the cortex of the adrenal glands to secrete aldosterone, which, by retaining sodium, maintains the correct intravascular volume. The feedback mechanism causes the kidneys to reduce the release of renin with an increase in blood pressure or an excess of sodium ions, which facilitates the excretion of a larger pool of sodium and thus leads to the normalization of blood pressure. Angiotensin II is responsible for the immediate control of short-term pressure, while the sodium-volume mechanism provides long-term control. Angiotensin II causes arterial vasoconstriction and the release of aldosterone and vasopressin [10,32,33].

### 1.3. Aldosterone

It is synthesized in the glomerular zone of the adrenal cortex and is induced by an increase in the level of the renin–angiotensin II system and potassium. Aldosterone synthesis is also influenced to a lesser extent by the adrenocortical hormone (ACTH) secreted from the anterior pituitary gland. The site of action of aldosterone is in the convoluted distal tubules of the kidneys and the collecting ducts of the nephrons [10,34]. Aldosterone influences sodium and water re-absorption and the excretion of potassium and hydrogen ions by the cells of the distal and collecting ducts. Re-absorption of sodium and water is considered to be the main mechanism of increasing blood pressure [34,35].

### 1.4. Electrolytes in the Male Reproductive System

Additional sex glands such as seminal vesicles, prostate gland, and bulbourethral glands, secrete substances that protect spermatozoa in the plasma of the semen, such as proteins, enzymes (acid phosphatase, alanine transaminase), lipids, electrolytes, and microelements- copper, iron, zinc [36,37]. The plasma of semen was found to be rich in electrolytes, such as calcium, sodium, and magnesium [38,39,40].

These elements maintain the osmotic balance of semen and are essential for metabolic processes, spermatogenesis, sperm maturation, as well as sperm capacitation and motility [41,42]. Variation in the values of Ca^2+^ and Mg^2+^ in seminal plasma is associated with infertility and poorer quality of semen quality [40,41,42,43]. During the research on roosters, semen was collected from 10 birds, which were assessed in terms of sperm viability and motility, calcium, magnesium, phosphorus, iron, and copper content [44]. The mean content for these examined elements were, respectively: 6.52 ± 0.4 mg/dL for Ca; 3.89 ± 0.18 mg/dL for P; 3.89 ± 0.2 mg/dL for Mg; 231.58 ± 6.23 μg/dL for Fe; and 425.40 ± 17.18 μg/dL for Cu. It was shown that the motility of a rooster’s sperm was significantly (*p* < 0.05) related with the content of calcium (r = 0.57), magnesium (r = 0.42), iron (r = 0.87), and copper (r = 0.66) in the semen plasma. Moreover, a significant (*p* < 0.05) relationship was found between sperm viability and the content of Ca (r = 0.58), Mg (r = 0.41), Fe (r = 0.87), and Cu (r = 0.67) in the semen plasma. These results indicate the influence of the tested element in the semen plasma on sperm motility and viability [44]. Sodium and potassium have been shown to be involved in the acrosomal reaction and are also associated with the normal functioning of sperm [40,41]. Umar et al. [45] carried out research on five mature and healthy bucks, from which the semen was collected twice a week for six weeks by the artificial vagina. This study aimed to establish a relationship between a buck’s semen quality and the levels of biochemical constituents of semen plasma. The mean volume of ejaculates was 1.19 ± 0.03 mL, an individual motility of sperm was at the level 89.18% ± 0.37%, a sperm concentration was 1.86 ± 0.04 × 10^9^/mL, and the content of electrolytes was Na^+^: 144.12 ± 1.59 mEq/L, K^+^: 27.38 ± 0.49 mEq/L, Cl^−^: 65.73 ± 0.45 mEq/L, Ca^2+^: 9.34 ± 0.22 mg/dL, P: 19.32 ± 0.97 mg/dL. There was a significant correlation between the parameters of semen quality, such as semen volume, sperm motility, and content of sperm, and the content of biochemical components of semen plasma. The volume of ejaculate showed a significant correlation (*p* < 0.01) with the content of Cl^−^ (0.355), and at the level of *p* < 0.05 with the content of K^+^ (0.233) and Ca^2+^ (0.190). There was also a relationship between sperm motility (*p* < 0.05) and sodium content (0.142). In turn, a negative correlation (*p* < 0.05) of sperm concentration was observed with calcium (−0.120) and phosphorus (−0.262). During this research it was found that most parameters of bucks semen quality were positively correlated with biochemical components [45]. There is also a correlation between the sodium and potassium content of the sperm plasma and the ability to fertilize. Infertile men have been shown to show low potassium levels compared to fertile men. These electrolytes are also related to sperm quality. There is a relationship between Na^+^ and K^+^ levels and sperm motility [40]. Moreover, sodium ions have been implicated in sperm capacitation, which is related to the membrane potential of sperm [3,40,46]. The Na, K-ATPase (NKA) is a unique ion transport enzyme in the plasma membrane that maintains the transmembrane concentration of sodium and potassium ions. To function properly, NKA needs the energy from ATP hydrolysis to catalyze the exchange of Na^+^ into K^+^. The ion exchange takes place through phosphorylation and dephosphorylation, which leads to conformational changes within the NKA structure, and the maintained concentration gradient of Na^+^ and K^+^ leads to the maintenance of the membrane potential. There are four isoforms of the NKA catalytic subunit-NKAα1, NKAα2, NKAα3, and NKAα4, but NKAα4 (ATP1A4) is only produced in male germ cells of the testis. It is known that NKAα4 is present in various species such as chimpanzee, gorilla, rabbit, rat, beaver, cheetah, lemur, camel, human, deer, bear, goat, sheep, bulls, and boars. Moreover, NKAα4 has a high affinity for sodium ions, and therefore it helps to maintain a low concentration of intracellular Na^+^ in sperm. It was proved that divalent cations such as Ca^2+^, Cu^2+^, Fe^2+^, and Zn^2+^ inhibit the activity of NKAα4, which indicates that they are not natural substrates for the enzyme. The NKAα4 isoform is present in sperm, in which it participates in their motility and hyperactivation. The ATP1A4 is involved in maintaining the resting and membrane potential of sperm, which is necessary for sperm motility and capacitation [47]. In the research on male mice with the Na, K-ATPaseα4 knockout is sterile and their sperm is not able to fertilize the oocyte also in vitro [48]. Sperm deficiency in α4 controlling ion homeostasis and the potential of the cell membrane results in flagellum bending, and improper ion regulation within the sperm, resulting in an increase in intracellular sodium levels and depolarization of the sperm membrane [48]. In turn, magnesium and calcium are key in sperm capacitation, acrosome reaction, and spermatogenesis, and affect sperm motility [39,40,41]. The sperm membrane is also involved in the capacitation process, during which the fluidity of the membrane is increased, and cholesterol is laterally displaced to the region of the sperm’s main apex. The ratio of cholesterol to phospholipids in sperm depends on the species of animals—in boars, it is 0.20, in bulls 0.40, in stallions 0.36. It has been shown that the higher this ratio is, the longer the incubation time is needed to achieve capacitation. Male sperm capacitation can be performed in vitro in media containing electrolytes such as Na^+^, K^+^, Cl^−^, HCO^−^_3_, Mg^2+^, Ca^2+^, PO^3−^_4_, energy substrates-glucose, pyruvate, lactate, as well as a cholesterol acceptor (serum albumin). In the activation of human sperm in vitro, they encounter higher concentrations of HCO^−^_3_, which stimulate ADCY10 adenylyl cyclase, the activation of which depends on two ions: Ca^2+^ and HCO^−^_3_. This increases the synthesis of cyclic adenosine monophosphate (cAMP). The cAMP levels are dynamic and regulated by ADCY10 and degradation by phosphodiesterases (PDE). The cAMP act on protein kinase A (PKA) closely related to sperm biology. The PKA consists of two catalytic and regulatory subunits. The regulatory subunit binds to the cAMP, while the catalytic one dissociates as an active kinase. The cAMP/PKA signaling pathway is necessary for human sperm capacitation and is mediated by the inactivation of HCO^−^_3_, Ca^2+^, Na^+^, and K^+^. These ions translate into the membrane potential and intracellular pH of the sperm. Regulation of the intracellular pH of the sperm occurs during the passage of gametes from the epididymis to the fertilization site, where the pH changes due to the presence of hydrogen ions (in humans, the vaginal pH is approx. 4.4, in the uterus approx. 7.0, and in the epididymis approx. 6.8). The pH is also influenced by HCO^−^_3_, which approx. 2–4 mM in the porcine epididymis, and approx. 20–60 mM in the female reproductive system. As sperm migrate through the female reproductive system to the site of fertilization, the sperm has an alkaline pH, which is necessary for capacitation and activation of CatSper channels. The alkalization of sperm is mainly based on the activity of the Na^+^/H^+^ (NHE) exchanger. During capacitation, an increase in the concentration of anions in the intracellular space is noted. In human sperm after hyperactivation, the membrane potential is about −58 mV, and before it is −17.7 mV. Another important ion is calcium. The Ca^2+^ has been shown to bind directly to membrane phospholipids and enzymes, leading to modification of membrane properties and enzymatic activation. Calcium can also bind to calmodulin (CaM), which causes conformational changes and modulates the activity of adenylyl cyclases, phosphatases, phosphodiesterases, and protein kinases. After ejaculation, the sperm are exposed to higher concentrations of HCO^−^_3_ and Ca^2+^, resulting in an increase in cAMP levels by the activity of ADCY10 and PDE, which in turn stimulate PKA-dependent protein phosphorylation. It has been shown that in human sperm, a decrease in calcium levels resulted in a decrease in ADCY10 and cAMP activity [49]. High levels of Ca^2+^ in sperm plasma have been shown to be associated with high levels of testosterone in Leydig cells [39]. Furthermore, calcium is associated with sperm physiology and quality, as well as metabolism [43]. Calcium and potassium have been shown to influence the movement of sperm flagella [43]. In the research conducted by Azab et al. (2021) [43] on 50 men divided into two groups—fertile (N = 20) and infertile with a varicocele (Vx) (N = 30), semen was assessed in terms of calcium and magnesium content and the quality of semen. The studies reported significantly lower levels of calcium (11.0 ± 2.9 mg/dL) and magnesium (5.0 ± 0.6 mg/dL) in infertile men before surgery of Vx compared to the fertile men (15.4 ± 2.6 mg/dL; 6.0 ± 0.7 mg/dL) [43]. Moreover, during the conducted research the highest motility of sperm was noted in fertile men (58.5 ± 4.9%) compared to infertile men before surgery Vx (20.6 ± 6.2%), and in turn, after the surgical removal of varicocele, sperm motility was at the level of 45.0 ± 12.1%, and the content of calcium 15.5 ± 3.7 mg/dL and magnesium 5.5 ± 0.6mg/dL. The percentage of normal sperm forms in fertile men was 6.7 ± 0.99% and for men with Vx = 2.1 ± 0.8%, on the other hand, after the surgery, the percentage of normal sperm was 5.9 ± 1.1%. The authors obtained a significant correlation between the level of Ca in semen and the levels ratio showed significant positive correlations with the concentrations of sperm (r = 0.479, *p* = 0.001; r = 0.541, *p* = 0.001; r = 0.282, *p* = 0.001), total sperm motility percentage (r = 0.493, *p* = 0.001; r = 0.477, *p* = 0.001; r = 0.353, *p* = 0.001), and the percentage of normal sperm forms (r = 0.578, *p* = 0.001; r = 0.520, *p* = 0.001; r = 0.430, *p* = 0.001, respectively) [43]. It has been shown that in infertile males, calcium and magnesium values are significantly lower than in fertile males. In addition, the levels of Ca and Mg in semen showed a positive correlation with sperm concentration, sperm motility, and normal morphology [43]. The research carried out by Venkata Krishnaiah et al. [50] investigated the effect of supplementation on the quality of goat sperm. The study included 40 goats divided into 10 groups, the control group was fed a basic diet, without additional supplementation, and the research groups received appropriate doses of zinc (20, 40, 60 mg of Zn), copper (12.5, 25, 37.5 mg Cu) and zinc along with copper (20 mg Zn + 12.5 mg Cu, 40 mg Zn + 25 mg Cu, 60 mg Zn + 37.5 mg Cu) for 26 weeks. To determine the effect of supplementation on semen and hormone levels, blood and semen were collected from animals. In the group of animals treated with 40 mg Zn (87.4 ± 1.3%), 12.5 mg Cu (87.75 ± 0.9%), 25 mg Cu (83.4 ± 1.8%), and 20 mg Zn + 12.5 mg Cu (87.7 ± 1.4%), 40 mg Zn + 25 mg Cu (85.4 ± 2.1%), and 60 mg Zn + 37.5 mg Cu (79.2 ± 2.9%) clearly increased (*p* < 0.001) sperm motility was noted compared to the control group (66.2 ± 5.0%). Moreover, the animals with a higher content of sperm in the ejaculate were 40 mg Zn (2258.4 ± 170.2 million/ejaculate), 12.5 mg Cu (2666.3 ± 310.6 million/ejaculate), 37.5 mg Cu (1880.6 ± 184.7 million/ejaculate), 25 mg Cu (2091.9 ± 172.6 million/ejaculate), 40 mg Zn + 25 mg Cu (2162.2 ± 228.9 million/ejaculate), and 60 mg Zn + 37.5 mg Cu (2418.8 ± 358.9 million/ejaculate) compared to the control group (1209.7 ± 129.3 million/ejaculate). On the other hand, in the case of hormones, a lower luteinizing hormone content was observed at week 32 in the groups treated with copper and the Zn and Cu combination (results in the range of 0.3–0.5 ng/mL) than in the control group (about 1.3 ng/mL). In the groups of 12.5 mg Cu, 37.5 mg Cu, 20 mg Zn + 12.5 mg Cu, and 40 mg Zn + 25 mg Cu, a lower testosterone content in the sperm plasma was obtained, amounting to about 3 ng/mL compared to the control (about five ng/mL). However, in the case of supplementation with 20 mg and 60 mg of zinc, higher testosterone contents (about five and 5.5 ng/mL, respectively) were obtained than in the control sample. The authors suggest that supplementation with zinc and copper, or a combination of Zn and Cu, may accelerate male maturation by about a month, and improve sperm quality [50].

### 1.5. Electrolytes in the Female Reproductive System

During female sexual cycles, changes occur in the regulation of fluid and electrolytes [6,51]. Both estrogen and progesterone, have been shown to influence the regulation of body fluids and hormonal fluctuations during menstruation in women. This causes changes in nutritional needs, sports performance, and temperature regulation, and these hormones influence the accumulation of interstitial fluid in the luteal phase [52]. Estrogen and progesterone have been found to affect the balance of water, and sodium content in the body, and influence the centers and hormones that control thirst [6,53]. Progesterone inhibits aldosterone-dependent sodium reabsorption in the distal tubules of the nephron and causes transient excessive sodium excretion (natriuresis) [6,51]. It may also compete with aldosterone for the mineralocorticoid receptor, leading to a reduction in the action of aldosterone or an increase in its synthesis, which is associated with an increase in sodium and water reabsorption in the kidney [6,51]. The natriuretic effect increases the concentration and activity of renin, angiotensin II, and aldosterone in the luteal phase, which also causes an increase in thirst. The renin–angiotensin–aldosterone system is activated [51]. In turn, estrogen increases blood volume by lowering the vasopressin release point and participates in the increase in the concentration of aldosterone in the luteal phase. Thus, it is concluded that sodium restriction in the diet in the premenstrual period may affect the synthesis of aldosterone in the luteal phase [6]. In turn, estrogen modulates the renin–angiotensin–aldosterone system by lowering aldosterone levels [53].

The concentrations of electrolytes such as Na^+^ and Cl^−^ have been shown to increase significantly on the day of ovulation. There is a correlation between sodium, potassium, and chloride levels and progesterone levels, as well as between estradiol and Cl^−^. According to Satué et al. (2021) [54], ovarian steroid hormones influence the excretion of mares. At the highest concentration of estrogen, loss of Na^+^ is observed, while at the beginning of the luteal phase in sheep, sodium reabsorption is observed, which is associated with the secretion of progesterone and aldosterone. Progesterone causes a decrease in sodium levels, as well as potassium in females. In turn, ovulation is followed by the highest K^+^ concentration in sheep, mares, and women. During ovulation, the excretion of potassium ions in the urine is reduced. A correlation of Cl^−^ with progesterone concentrations has also been demonstrated in mares and women [54]. In the research conducted by Satué et al. (2021) [54] in healthy mares aged about seven years, it was shown that the highest concentration of sodium and chloride were recorded during ovulation (day zero) compared to the pre- and post-ovulation period. The concentration of Na^+^ was 142.2 ± 3.28 nmol/L and for Cl^−^: 107.4 ± 1.44 nmol/L. In turn, the highest potassium concentration was recorded the day after ovulation—4.51 ± 0.57 nmol/L [54]. In cows, a correlation of Cl^−^ was obtained with the diameter of the estrogen follicle and the concentration in the buffaloes [55]. In research conducted on women, a decrease in electrolyte concentrations, such as Na^+^ and Cl^−^, was observed in the luteal phase compared to ovulation [51]. Furthermore, during ovulation, there was an increase in electrolyte content in the plasma, higher thirst in the examined women, and an increase in estrogens. The osmoregulatory effect of estrogen is associated with an increase in plasma volume. In the research carried out on women at various stages of the menstrual cycle, blood samples were collected, from which the concentration of electrolytes in the plasma was analyzed. The highest concentration of Na^+^ (143.8 ± 2.81 mmol/L) and Cl^−^ (110.7 ± 1.85 mmol/L) during the ovulation was obtained, while lower values were noted in the late follicular phase (Na^+^: 139.4 ± 1.30 mmol/L; Cl^−^: 105.8 ± 0.63 mmol/L). There were no statistically significant difference between the concentration of potassium depending on the phase of the cycle. The content of K^+^ was approximately 4.0 mmol/L. The authors attribute the increased chloride and sodium plasma concentration of women during the ovulation to increased estrogen-related salt retention. In turn, a lower content of these two electrolytes in the luteal phase may be related to the natriuretic action of progesterone [51]. This increase can be attributed to estrogen and aldosterone-dependent resorption [51].

Ca^2+^ and Na^+^ play an important role in the production of lipid hormones in developing follicles and regulate the secretion of the breeding hormones necessary for ovaries and ovulation [54]. Calcium is an important electrolyte in reproduction. It plays an important role in the production of developing follicle hormones, as well as in the regulation of the secretion of hormones necessary for ovaries and ovulation [56]. Calcium ions are also important in estrogen synthesis, the levels of which increase during follicle development, necessitating the accumulation of Ca^2+^ in the follicular fluid. Calcium is involved in the meiotic cell cycles of mammalian oocytes [57]. In research conducted on birds, it has been found, that calcium may also play a role in increasing follicle growth [58]. It was found that Ca^2+^ deficiency in the diet of laying birds reduced egg production (approximately 47% of Ca-adequate control diets containing 3.6% Ca). Moreover, the effect of calcium deficiency on the number of hierarchical follicles was limited (approximately 56% of Ca-adequate control diets containing 3.6% Ca). The authors also report that calcium deficiency in laying birds significantly limits the process of selecting pre-ovulatory follicles [58]. Moreover, the development of ovarian follicles in birds is orderly. The avian granulosa cell layer is important during follicle differentiation where follicle-stimulating hormone receptor (FSHR) is expressed in the granulosa cells. The FSHR expression is maintained by bone morphogenetic protein (BMP) 4 and BMP6, transforming growth factor β (TGF β), and activin. Before follicle selection, the differentiation of the granulosa cell layer in birds is suppressed by active mitogen-activated protein kinase (MAPK) signaling through extracellular signal-regulated kinases (ERK 1, ERK 2) [59]. Undifferentiated granulosa cells express low levels of steroidogenic acute regulatory protein (STAR), luteinizing hormone receptor (LHR), cytochrome P450scc (CYP11A1), and follicle-stimulating hormone receptor (FSHR), and during selection, STAR and CYP11A1 expression and follicle-stimulating hormone secretion increase, and follicles begin to produce progesterone [60]. Chen et al. [58] hypothesized that calcium influences follicle selection via the signaling pathway of protein kinase A (PKA)/cAMP-mediated mitogen-activated kinase (MAPK) along with gonadotropic hormones. To confirm the hypothesis, Chen et al. [58] conducted a study on 450 ducks fed with the poor feed, and then with one rich in calcium. The animals were divided into three groups depending on the amount of calcium in the diet—3.6% calcium (control, appropriate amount), 1.8% Ca, and 0.36% calcium. The animals were initially kept on diets for 67 days, and then for the next 67 days, they were fed ab libitum with the control feed. During the study, there was a reduction in egg production with a low calcium diet (egg weight was reduced by approximately 52% for the 0.36% calcium diet) compared to the control. Moreover, in the case of the lowest dose of calcium in the diet, a decrease in the weight of the ovary and the number of hierarchical follicles was observed. In turn, higher levels of a MP in the ovaries were noted in both diets of ducks deficient in calcium, which were then reduced during the treatment of calcium deficiency. cAMP plays an important role in the selection and maturation of follicles in birds, in which the follicle-stimulating hormone is also involved. cAMP influences the activation of the PKA protein and then phosphorylates MAPK and is associated with the arrest of oocyte meiosis. It was shown that higher PKA phosphorylation was obtained in the group of ducks fed on a low-calcium diet, which was caused by the accumulation of calcium in the duck ovaries, which then suppressed the selection of follicles. The authors showed that the effects were reversible-by giving the ducks the right dose of calcium [58]. The Ca^2+^ level has also been found to increase after the release of the luteinizing hormone [57]. Intracellular calcium growth is the basis for oocyte activation and fertilization [61,62,63]. The sperm-specific phospholipase C zeta (PLC-ζ) is believed to be the necessary stimulus for the generation of calcium oscillations, which then activates oocytes and early embryonic development [64]. After the fusion of sperm membranes with oocytes, there is a rapid increase in calcium content and resumption of meiosis II [65]. The PLC-ζ induces the release of Ca^2+^ through the inositol 1,4,5-triphosphate (IP3) signaling pathway via the hydrolysis of phosphatidylinositol 4,5-bisphosophate (PIP2) [64]. Channels for Cl^−^, K^+,^ and Ca^2+^ ions also participate in the fertilization and activation of oocytes. During activation of murine oocytes, the membrane potential changes, during which the permeability of the membrane to K^+^ and Na^+^ is noted. During activation, there is an influx of calcium ions into the endoplasmic reticulum via the IP_3_ receptor (IP_3_R) or the ryanodine receptors (RyR), which have been localized, for example, in rodents. Frog oocytes are a model for studying oocyte activation processes. It has been shown that after sperm fusion with the oocyte, calcium-activated chloride channels (CaCC) are activated, which causes membrane depolarization, counteracting polyspermy. The Ca^2+^ then induces a calmodulin protein kinase II (CaMKII) to initiate activation of the oocytes, followed by phosphorylation of Emi2, a component of the cerebrospinal fluid contributing to the recovery of oocyte anaphase II. This then leads to the activation of the anaphase-promoting complex (APC), which contributes to the termination of the meiosis [61]. In research conducted by Wang et al. [66], the effect of various calcium transporters inhibitors, including erastin, and a T-type calcium channel inhibitor, on mouse oocyte were tested. It was established that the Ca^2+^ in the Chatot-Ziomek-Bavister (CZB) culture medium is an essential factor for the maturation of oocytes. It has been shown that calcium channels are important in this process, and their inhibitor blocks the disintegration of germinal vesicles. It was also shown that oocytes cultured on the CZB medium without calcium, died within two hours of the disintegration of the germinal vesicle, indicating the need for calcium in the environment [66]. Oocytes can fertilize after an activation process, in which sperm is involved. The presence of sperm causes an intracellular oscillation in calcium concentrations that triggers a cascade of reactions such as cortical granule exocytosis, resumption of oocyte meiosis, and prevention of polyspermy [67,68]. Sodium is associated with follicle viability and its activity in estrogen synthesis. In studies carried out on the ovaries of 50 camels, significant differences (*p* < 0.05) were found between the size of the follicle and the concentration of Na^+^ and K^+^ [56]. In small follicles, 3–9 mm in size, the sodium concentration was 93.33 ± 4.75 mmol/L, while in large follicles (10–19 mm)—145.96 ± 4.26 mmol/L. Moreover, the follicle’s size increased as they continued to grow by osmosis into the follicular fluid. Similar results were obtained in studies on goats and sheep. The concentration of K^+^ also differed significantly (*p* < 0.05). In small follicles, it was 12.96 ± 0.68 mmol/L, and in large follicles, it decreased and amounted to 6.12 ± 0.57 mmol/L. The reduction in the content of these ions is associated with the development of the follicle [56]. The decrease in potassium concentration is associated with the development of the follicle, which leads to the transfer of potassium ions from the extracellular space to the intracellular space, and thus reduces its concentration in the follicular fluid as the follicle becomes larger. The concentration of K^+^ in the follicular fluid was of great importance compared to its concentration in the serum, with no correlation between them, indicating that the potassium ion could be excreted locally in the follicular fluid. In general, this study validated the variability in the concentration of metabolic components and ionic follicular fluids depending on the size of the follicle and its stage of development. The results of this study can be considered [56]. Potassium channels are found in the ovary and the myometrium of the uterus, which is involved in the secretion of progesterone. These channels play an important role in the cellular response depending on the concentration of Ca^2+^. The myometrium contains potassium channels that are sensitive to calcium ions. These channels play a role in the regulation of uterine contractility. Calcium-activated potassium channels are involved in inducing uterine smooth muscle relaxation by mediating membrane depolarization. During pregnancy, the level of calcium increases. Sodium, potassium, and calcium ions, as well as ion channels, are involved in the contraction and relaxation of the uterus [69].

Table 5 shows the role of selected electrolytes in the male and female reproductive systems.

## 2. Electrolyte Disturbances

Maintaining electrolyte homeostasis depends on kidney function and steroid hormones [9,10,20]. The kidneys filter the body of toxins circulating in the bloodstream, which directly regulates electrolyte balance. The pathological states of this organ cause a disturbance of the entire water and electrolytes [9,28]. Kidney disease and dysfunction (chronic kidney disease—CKD) result in the impaired regulatory function of the kidneys, which can lead to metabolic acidosis, alkalosis, hyperkalemia, hyponatremia, hypercalcemia, and hyperphosphatemia [9,20]. Pathologies and abnormalities in the animal’s body can cause damage to some key internal organs due to an imbalance or instability. Disruption of the metabolic processes that regulate the basic vital functions of an animal can lead to enormous health complications, such as bone mineralization disorders, vascular disorders, calcification, and even mortality [28,73,74].

Acidosis occurs mainly due to a decrease in renal acid excretion with continued metabolic acid production, resulting in acid retention and a drop in blood pH below 7.35. Acidosis symptoms are mainly headache, sleepiness, coma, dry cough, shortness of breath, tachycardia, arrhythmia, weakness, diarrhea, nausea, and vomiting [9,75]. Alkalosis occurs when there is a loss of acidity or an increase in alkalinity by extracellular fluid, then no renal excretion of HCO_3_^−^ is observed, and the pH is above 7.45. Symptoms are mainly headache, coma, hand tremor, arrhythmias, decreased contractility, twitching, muscle spasm, nausea, and vomiting [76].

Potassium is the most abundant intracellular cation that is important in excitable tissues such as the heart, nerves, and skeletal muscles as it is essential for action potentials and electrical excitability. Under physiological conditions, potassium is filtered by the glomeruli and the distal nephron is the major site of this ion regulation in the kidney [77]. Acid-base disorders affect potassium balance—acidosis decreases, and alkalosis increases potassium secretion by the kidneys. Hyperkalemia is the most common disturbance in the electrolyte balance resulting from renal dysfunction resulting from impaired water regulation. The most common causes of hyperkalemia are impaired glomerular filtration and poor potassium secretion [20]. Potassium levels above 5.3 mEq/L are defined as hyperkalemia. The most common symptoms are muscle weakness, muscle paralysis, and cardiac arrhythmias. In turn, hypokalemia (<3.5 mEq/L) occurs less frequently and is mainly due to alkalosis, the use of diuretics, vomiting, or diarrhea. Symptoms are muscle weakness and cramps, respiratory failure [9,77]. Sodium is an important factor that influences serum osmolarity, which is controlled by vasopressin [3]. Another disturbance in sodium metabolism is hyponatremia, which causes a decrease in serum sodium concentration (below 135 mEq/L), which is associated with dizziness, fatigue, lethargy, convulsions, headache, or malaise. Hypernatremia is associated with a sodium concentration over 145 mEq/L, and its main symptoms are thirst, muscle weakness, and confusion [46,78]. The second most abundant intracellular cation is magnesium, which is a cofactor of many enzymes and participates in DNA synthesis, replication, and phosphorylation of signaling proteins. Magnesium is mainly absorbed in the small intestine and to some extent in the large intestine. Both hypermagnesemia (>2.3 mg/dL) and hypomagnesaemia (<1.7 mg/dL) can manifest as muscle weakness, paresthesia, tremors, and convulsions [20,79].

The most common causes of water and electrolyte disturbances are presented in Figure 1.

Electrolyte disturbances affect fundamental systems in the body, so it is important to quickly identify the problem, and verify which specific ions have been disturbed. The too low level of the element is supplemented and when the upper limit of the norm is exceeded, it should be removed from the organism [46].

## 3. The Influence of Electrolyte Disturbances on Animals

Proper acid-base balance (for mammals the variation of pH is between pH 7.36 and 7.44) and electrolyte balance are essential to maintain body homeostasis [80]. The body eliminates changes in acid-base balance with the help of buffering, regulation of chemical buffering, respiratory adjustment of blood carbonic acid, and excretion of pH or HCO_3_^−^ by the kidneys [80]. Regulation of acid-base balance requires three organs: liver, lungs, and the most important—kidneys. Kidney diseases manifest themselves as gastrointestinal, cardiovascular, respiratory, hematologic, and nervous disorders [81].

Acid-base and electrolyte imbalances are common in cattle. In research conducted by Garzon-Audor et al. (2020) [82] in cows, it was observed that in calves younger than three months of age, metabolic acidosis is more common, while above three months of age, calves were more likely to be develop metabolic alkalosis. This may be due to changes related to the physiological development of the rumen in the first months of calves’ lives. Acidosis may be caused by neonatal diarrhea, which causes a loss of bicarbonate ions, and dehydrated calves have reduced renal excretion of hydrogen ions. On the other hand, in cows, metabolic alkalosis is diagnosed with disorders such as displacement and/or torsion of the abomasum and intestinal obstruction [82].

In cows experiencing heat stress, milk yield decreases; moreover, heat stress reduces the conception rate in dairy cattle, and may also have a negative impact on the success of fertilization and embryo mortality. Heat stress also reduces the immune value of cow colostrum and reduces the fat and protein content of cow milk of cows exposed to heat stress [83]. The animals may also die (Lethal Heat Stress-LHS). Disorders that progress through LHS are electrolyte disturbances and acid-base balance. At first, animals physiologically try to cope with heat stress by increasing the rate of breathing and sweating [64,84]. The consequence of this is electrolyte disturbances, which are manifested by a large amount of potassium ions released percutaneously. In addition, during heat stress, Na^+^ is excreted through the skin. Heat-stressed cows excrete potassium with sweat and are characterized by a lower level of aldosterone, which promotes increased reabsorption of K^+^ in the kidneys and the excretion of Na^+^ by the kidneys. Low levels of aldosterone in cows exposed to heat stress cause an increase in plasma volume due to increased water consumption [84]. Increased respiration rate helps animals cool down and reduces the concentration of CO_2_ in the blood, increasing the blood pH and resulting in respiratory alkalosis. Respiratory alkalosis stimulates HCO3^−^ excretion by the kidneys to maintain a stable blood pH, and the loss of bicarbonate ions results in metabolic acidosis [82,85]. Furthermore, a reduced concentration of aldosterone as a result of heat stress in cows leads to the retention of hydrogen ions in the kidneys, which favors the development of metabolic acidosis [84].

A study on cats has shown that abnormal sodium levels in the plasma of these animals are associated with death [74]. In animals with sodium values up to 162 mmol/L, lower mortality was observed than in the whole population. On the other hand, cats with concentrations below 150 mmol/L or above 162 mmol/L sodium had a higher mortality rate compared to the population. This was independent of the severity of the disease and indicated that most likely anomalies in the water balance had occurred earlier. The chloride concentration significantly affects the acid-base balance. Their increased levels lead to acidosis, whereas their decreased levels lead to alkalosis. Identifying the problem related to the concentration of these ions helps us to quickly assess the health of the animal. In the investigation, hyperkalemic cats showed mortality in animals with renal disease or urinary obstruction. Hypercalcemia has been correlated with falls, especially in cats diagnosed with cancer. Hypocalcemia has been observed in people with pancreatitis, which has been associated with sepsis and death over time [74].

The circulatory system and heart function can be significantly disturbed due to electrolyte imbalance. It was shown that in the case of dehydrated animals, that is, with a disturbed electrolyte balance, the heart rate increased and there were disturbances in the rhythm of this organ [86]. Such disturbances are often observed in calves with diarrhea [87]. Furthermore, there is an increase in the number of lymphocytes and eosinophilia, which may already indicate inflammation in the body [71]. Dehydration of calves was also accompanied by a decrease in the concentration of sodium and magnesium ions, while the concentration of potassium increased [71].

## 4. Reproductive Consequences Resulting from Disturbances in Electrolyte Balance in Animals

Electrolyte imbalance in animals affects reproductive performance and thus the continuity of breeding. It has been shown that acid-base imbalance affects the capacitation of sperm [88]. Disorders play an important role in the proper delivery and contribute to the health problems of newborns. This means an increase in the percentage of deaths and severe diseases that exclude the animal from further use. For this reason, it is important to detect any disturbances in the concentration of elements quickly and prevent them. Interpreting abnormalities may help to improve breeding indexes that have deteriorated as a result of water and electrolyte disturbances [89]. The influence of dietary electrolyte balance (dEB) on reproductive performance in 80 sows was investigated. The animals were divided into two groups in terms of values of dietary electrolyte balance in feed—165 and 300 mEq/kg. The pigs were fed with this diet from day zero of gestation to farrowing. All diets contained calcium, phosphorus, and potassium, while the levels of sodium and chloride were different to obtain a suitable dEB. Sows fed with the 165 mEq/kg during gestation were characterized by a higher (*p* < 0.05) total number of piglets born (12.2), born alive (11.6), totally weaned (10.7%), weaning litter weight (78.9 kg) and average birth weight (1.31 kg) than those fed the 300 mEq/kg diet (respectively: 10.6, 10.0, 9.6%, 67.9 kg, 1.30 kg). The influence of this disorder on the morphological profile was also checked. The change in dEB was correlated with a reduction in litter size at birth and weaning. This disorder also affected the body weight [89].

Electrolyte disturbances can threaten the proper process of animal reproduction. However, based on cattle, researchers determined that the continuation of research is necessary to refine the correlation of abnormalities in this area to prevent the emergence of further breeding problems more effectively. Disruptions in the cellular stage can exclude both the cow and the bull from the reproductive process and consequently inactivate these animals in the context of breeding. Inhibition of reproductive cells is an obvious consequence of disturbances in the electrolyte-hormone line. As a result of anomalies in the feedback line, reproductive processes become ineffective, and the animal’s health deteriorates. Thus, its welfare and breeding suitability are reduced. It is necessary to continue discussions on this matter to explore known issues and specify the possibilities of preventing such situations in advance based on the electrolyte-reproductive prevention [90]. The way to regulate the concentrations of individual elements in the body is to supplement them in the diet. It is a proven method of maximizing reproductive performance while being the simplest way to improve health. Unfortunately, in the case of other causes of electrolyte disturbances, not due to dietary deficiency, the problem is more complex and requires urgent veterinary consultation, as it can potentially affect many systems and organs. Therefore, the issue of proper electrolyte management is very important in the context of good and effective animal care [91].

### 4.1. Electrolytes Disturbances in the Male Reproductive System

Currently, the causes of male fertility disorders are seen as testicular dysfunctions, testicular cancer, obesity, endocrine disorders, the impact of pesticides, exposure to heavy metals or oxidative stress, as well as environmental factors. There is a significant impact of nutrition, especially certain elements such as calcium, potassium, sodium, and magnesium, on the quality of semen [41,72].

In the male reproductive system, balance is necessary to maintain adequate sperm production and reproductive potential [92]. Spermatozoa after spermatogenesis are functionally immature and immotile. While moving through the epididymis, the sperm acquire fertilization competencies [93]. These changes depend on several proteins, and enzymes, but also on pH regulation related to ionic composition [92]. Homeostasis of the pH of the fluid in the lumen of the luminal fluid pH is essential for the process of spermatogenesis, maturation, and capacitation of sperm and thus for fertilization. Seminal tubular fluid (STF) has an acidic pH of 7.3 compared to plasma (approximately pH 7.5). To maintain the pH at this level, it is necessary to resorb Cl^−^, Na^+^, and HCO_3_^−^ with the simultaneous release of potassium ions [92].

The impairment of the ion channels in sperm results in male infertility. The main anion is Cl^−^, which plays an important role in semen, including being involved in sperm capacitation and the acrosomal reaction. Chloride channel blockers, for example, 4,4’-diisothiocyanatostilbene-2,2’-disulfonic acid (DIDS), have been shown to inhibit the acrosome reaction [72]. Disturbances in chloride ion concentration lead to the inhibition of sperm migration in the uterus [72]. Studies in guinea pigs have shown that incubation of sperm in the medium with chloride channel inhibitors leads to capacitation disorders. Moreover, disturbances in Cl^−^ concentration of Cl^−^ lead to disturbances in the sperm capacitation [94].

One of the most common causes of male infertility is asthenozoospermia, which is associated with a pathological reduction in sperm motility. Defects in transmembrane ion channels have been shown to cause asthenozoospermia [95]. Associated with the sperm cation channel (CatSper) is a protein channel located on the sperm plasma membrane. It is important in the process of sperm capacitation [95]. CatSper enables the influx of Ca^2+^ ions into the sperm flagellum during capacitation and the mobility of the gametes [96,97]. CatSper can conduct ions such as sodium; however, the affinity of calcium is much higher [98]. Ca^2+^ signaling is a conserved mechanism to modulate cell motility by increasing the asymmetry of the flagellum. The influx of Ca^2+^ is necessary to change the shape of the flagellum wave [99]. It depends on cAMP and pH and is activated by progesterone. Genetic defects in CatSper have been shown to reduce sperm motility, reducing male fertility [95,100]. Furthermore, CatSper delivers these ions to the sperm flagellum as the main calcium channel of sperm [96]. Elevated calcium levels are necessary for sperm motility [92]. Calcium channels are also crucial during semen migration through the female reproductive system [100]. Studies in mice have shown that CatSper is needed to pass from the uterus to the fallopian tube. [100]. Spermatozoa showing disorders within CatSper are unable to undergo the capacitation process and penetrate the zona pellucida, and thus cannot fertilize the oocyte [96]. The inactivity of CatSper is strongly associated with infertility [101].

Potassium channels are present in spermatogenic cells and the structures of mature spermatozoa. These channels are classified into voltage-gated potassium (Kv) channels, calcium-activated potassium (KCa) channels, internal rectifying potassium channels (Kir), and potassium channels in the tandem pore domain (K2P) [102]. They participate in the regulation of membrane potential and support sperm motility during the capacitation [103]. They are also involved in hyperpolarization, regulation of sperm volume, vital functions, and acrosome response in sperm [104]. Potassium channels play a role in maintaining hyperpolarization of the sperm membrane during capacitation, as well as in the maintenance of sperm volume [68,105]. The potassium channels described most frequently in sperm are Slo1 and Slo3, which belong to voltage-gated channels encoded by genes from the *SLO* family [97]. These channels are activated by depolarizing the membrane with calcium and magnesium. The increased potassium ion content in the extracellular environment induces an increased Ca^2+^ level, and in the case of human sperm, it is accompanied by the addition of progesterone [97]. Studies in mice have confirmed that the Slo3 subunit of the potassium channel (KCNU1) is essential for fertility, and any disturbance within the channel cause infertility in mice [100]. In a study by Zeng et al. (2011), [106] showed that depriving mice of Slo3 abolishes potassium conductance in mouse sperm, resulting in significantly reduced sperm motility and animal sterility.

Voltage-gated sodium channels (NaV) have been implicated in the regulation of sperm viability, motility, and their effect on membrane integrity and potential [68,103,104]. During sperm capacitation, the influx of sodium ions into the sperm occurs, and the opening of NaV channels causes increased motility, capacitation, and the acrosomal reaction of the bull sperm [97,107,108]. Chauhan et al. (2018) [107] studied the effect of veratridine on bull spermatozoa. Veratridine is an antagonist of voltage-gated sodium channels, which causes the activation and opening of sodium channels, also promotes the opening of CatSper calcium channels and leads to an increase in the concentration of this electrolyte in the intracellular space [107]. In research on bull spermatozoa incubated with concentrations of six, eight, and 10 μM veratridine, a time-dependent increase in spermatozoa motility was obtained. However, at higher concentrations of veratridine (eight and 10 μM), hyperactivity was obtained, which disappeared after two hours of incubation, and the sperm cells lost osmotic resistance and showed a bent neck condition. It may indicate the harmfulness of veratridine or mechanisms changing the ionic balance of sperm, which may be associated with the persistent opening of NaV channels [107]. More research is needed to understand the interactions between Nav and other channels in regulating the capacitation process and basic mechanisms in sperm.

Malfunctioning of these ion channels can affect sperm functions such as over-activation, capacitation, and the acrosome response, which will affect fertilization and then may lead to infertility [105,107]. The loss of electrolyte balance results in the loss of continuity of the sperm membrane, leading to hypoosmotic edema [97].

Electrolyte disturbances also lead to impaired functioning of entire systems as a result of abnormalities in cell metabolism. It has been proven that a change in pH and a disturbance in acid-base balance led to stress, dehydration, and inhibition of reproductive processes, such as sperm production, even within six days. Such changes can occur as a result of lack of food, and water, changes in the external environment, and animal disease [109]. Electrolyte disturbances can take place in the laboratory, but urine and feces can also be evaluated [110]. The presence of mucus or blood is especially noteworthy. The nature of bowel movement and its frequency are important. The reabsorption of sodium and water serves to dehydrate the feces before defecation. Perturbations in this area can cause the animal diarrhea and aggravation of electrolyte imbalance, and lead to a systemic inflammatory response [111].

### 4.2. Electrolyte Disturbances in Female’s Reproductive System

Follicular fluid (FF) is the plasma filter, and the secretion of follicular cells contains a wide variety of components, including proteins, hormones, and exosomes. It participates in the communication between the oocyte and the cumulus cells, which influences the maturation of the oocytes [112]. FF can serve as an indicator of the functional status of the ovarian follicle. However, the literature does not describe the collection of follicular fluid to assess the acid-base balance from live animals. Maintaining the acid-base balance in body fluids is essential for maintaining whole-body homeostasis, and many deviations indicate disorders [113]. Female fertility disorders are correlated with electrolyte concentration [66]. The composition of the follicular fluid changes as the follicle grows. The bovine follicular fluid contains ions such as magnesium (Mg^2+^), chloride (Cl^−^), sodium (Na^+^), and potassium (K^+^). The composition of FF is also influenced by the stage of lactation, uterine diseases, or the poor quality of cow oocytes. In the research conducted on heifers (N = 10), lactating cows (N = 8), and cows from repeat breeding (RB, N = 11), and the levels of sodium, potassium, chloride in follicular fluid were measured [114]. It has been noted that average concentrations of sodium and potassium in FF in heifers (Na: 139.87 ± 1.83 mEq/L; K: 3.96 ± 0.12 mEq/L) were lower (*p* < 0.05) than in cows in lactation (Na: 138.60 ± 1.75 mEq/L; K: 4.00 ± 2.06 mEq/L) and in RB (Na: 140.72 ± 1.79 mEq/L; K: 4.35 ± 0.46 mEq/L). The chloride concentration in lactating cows was higher (120.80 ± 3.40 mEq/L) than in heifers (97.90 ± 3.73 mEq/L) and in RB cows (113.00 ± 11.27 mEq/L) [114]. The Cl^−^ ions initiate luteinizing hormone-stimulated steroidogenesis in chicken granular cells, amphibian oocytes, and steroidogenesis in the adrenal glands of rats. The lower concentration of Cl^−^ in the ovulatory follicles of lactating cows may be due to lower steroidogenesis in lactating cows compared to dairy heifers during the estrus [114].

Partial or complete fertilization failure (TFF) can be associated with a lack of calcium oscillation [65]. Fluctuations in the calcium profile can have a negative impact on the fertilization process and embryonic development [52,56,100]. The oocyte activation deficiency (OAD) may result from levels of PLC-ζ that are too low to cause calcium oscillation. Therefore, calcium influx stimulants such as calcium ionophore, calcimycin, and calcium chloride are used in patients during assisted reproduction to artificially induce oocyte activation (AOA) [115]. Studies on 178 patients showed an increased rate of fertilization (approximately 29% ICSI vs. approximately 50% ICSI-Ca) by calcium supplementation to ICSI media (intracytoplasmic semen injection) compared with the non-supplementation group. The studies improved fertilization, implantation, as well as the pregnancy rate using the ICSI method with Ca^2+^ supplementation. The authors suggest that calcium supplementation of sperm before injection into the oocyte may also affect the performance of sperm, which need calcium ions for the capacitation process and the acrosome reaction [115]. Insufficient calcium content may limit the implantation capacity, while too high a Ca^2+^ concentration may result in impaired development after implantation. Abnormalities in Ca^2+^ content and oscillations may result from the atypical expression of calcium-related protein genes. It is known that abnormal calcium ion concentration values after conception can interfere with gene expression by modifying the blastocyst transcriptome. During oocyte activation, calcium causes the production of reactive oxygen species (ROS) that restart the oocyte cell cycle [48,56,57]. On the other hand, in the case of fluctuations in calcium levels, ROS overproduction can occur, which translates into the induction of oxidative stress within the oocytes. Oxidative stress can promote apoptosis, and methylate oocyte DNA, which may translate into the failure of fertilization [65]. In addition, oocyte mitochondria contain calcium channels that are important in regulating calcium influx and influx. Damage to the mitochondria in oocytes has been shown to reduce female fertility due to the lack of activation of oocytes [66].

Potassium channels, as well as intracellular potassium concentration in the ovary, are involved in the regulation of progesterone secretion. Blocking or disturbance of potassium channels leads to a decrease in progesterone secretion [69]. According to Liu et al. (2020) [116], K^+^ deficiency affects hormone synthesis. Potassium deficiency has been shown to decrease progesterone levels with a slight increase in estrogen, which can translate into a decrease in levels of gonadotropin-releasing hormones associated with ovarian weight loss. As a consequence, it leads to reproductive disorders in women [116].

Ion channels have been shown to influence the functioning of the female reproductive system and therefore have an impact on fertility [117,118]. The potassium channel plays a key role. Mutations in the gene that encodes the *KCNQ1* K + potassium channel are associated with female infertility. The voltage-gated potassium channel *KCNQ1* has been detected in the endometrium of the uterus, where it is involved in the regulation of the repolarization of the uterine action potential. Furthermore, mutations in this gene are associated with endometriosis, inhibiting the degradation of catecholamines such as adrenaline, dopamine, and noradrenaline, thus increasing the level of stress hormones leading to inhibition of the hypothalamic–pituitary–ovarian axis. This can reduce fertility as well as interfere with gamete transport of gametes through the fallopian tube [117]. Moreover, ion channels have been detected in the uterine endometrium of many species (mice, rats, humans, pigs) that are involved in the regulation of endometrial receptivity and embryo implantation [118].

A cAMP-activated anion channel of CFTR was detected in the endometrium. It is involved in the outflow of Cl^−^. In humans, mutations in the CFTR gene cause cystic fibrosis, which is characterized by defective electrolyte and fluid transport in many different epithelia and has been long observed with fertility problems in both men and women. CFTR is considered involved in the implantation of the embryo in the uterus [118]. Research by Ling et al. (2020) [119] showed that fertile women expressed the expression of intracellular chloride channel 4 (CLIC4) in the endometrium, while women with unexplained infertility did not have this expression [119]. CLIC4 has been shown to regulate the absorbency and facilitate blastocyst attachment of the blastocyst to initiate implantation. Decreased CLIC4 levels may be the cause of implant failure in women [105].

## 5. Conclusions

Electrolyte balance is very significant, ensures homeostasis of the organism, maintains reactions, nerve conduction, and proper polarity of cell membranes, and electrolytes are important in fertility. Male and female fertility is influenced by electrolytes. Electrolytes that have an excess of, or are deficient in electrolytes result in reproductive disorders. In the case of electrolyte disorders in the body, a lack of oocyte activation, altered endometrial embryo implantation in the endometrium, decreased sperm motility, decreased capacitation, or acrosomal reaction are noted.

## Figures and Tables

**Figure 1 biology-11-01006-f001:**
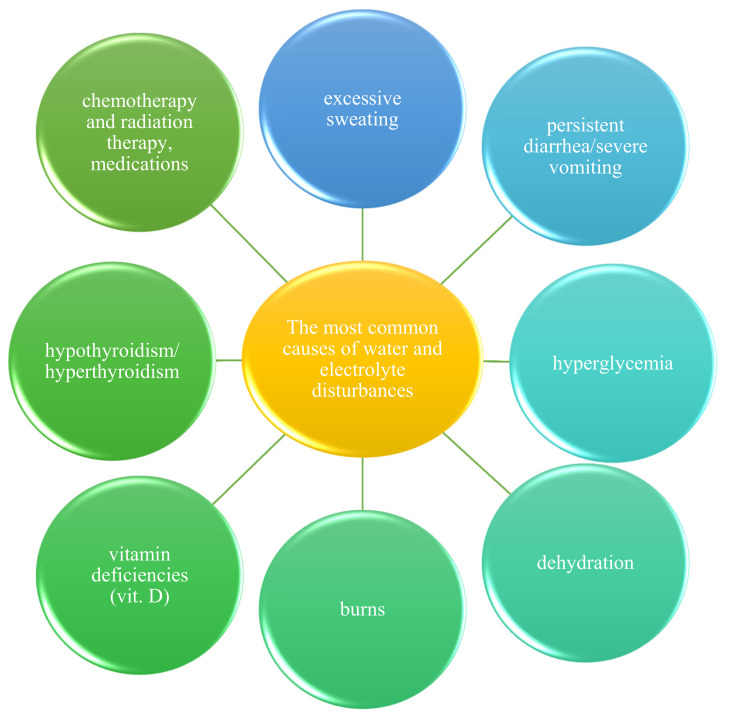
The most common causes of electrolyte and water disturbances; source: [46].

**Table 1 biology-11-01006-t001:** The most significant electrolytes in animal organisms.

Electrolyte	Functions	Regulation	Disorders	Literatures
**Sodium (Na^+^)**	Maintaining the volume of extracellular fluid, regulating of the membrane potential of cells, muscle contraction	Sodium re-absorption takes place in the proximal tubule and distal kidneys	Hyponatraemia-low serum sodium levels- headache, confusion, nausea, delirium. Hypernatremia-high serum sodium concentration-symptoms: rapid breathing, difficulty sleeping, and feeling anxious	[1,3,4,5]
**Potassium (K^+^)**	Establish the resting membrane potential in neurons and muscle fibers after membrane depolarization and action potentials. The sodium-potassium adenosine triphosphatase pump has the primary responsibility for regulating the homeostasis between sodium and potassium, nerve conduction	In kidneys, the potassium filtration takes place at the glomerulus under the influence of aldosterone	Cardiac arrhythmias, muscle cramps, muscle weakness, rhabdomyolysis, myoglobinuria are signs and symptoms in hyperkalemia	[1,3,6,7]
**Chloride (Cl^−^)**	Mostly in the extracellular fluid. Maintains serum electroneutrality, a key electrolyte for maintaining the acid-base balance, contributes to electrical activity (e.g., muscular and myocardial activities), contributes to the production of hydrochloric acid, secretion of fluids in the digestive tract, affects the transport of oxygen and gas exchange, contributes to the maintenance of blood pressure, affects the functions of the kidneys	The kidneys predominantly regulate serum chloride levels	Hyperchloremia can occur due to loss as vomiting or excess water gain, such as congestive heart failure	[3,8,9]
**Calcium (Ca^2+^)**	It is present in the extracellular fluid. Involved in skeletal mineralization, muscle contraction, nerve impulses, blood clotting, fertilization, and secretion of hormones	Parathyroid hormone (PTH) and calcitonin (CT) participate in the regulation of calcium. Absorption of calcium in the intestine is primarily under the control of the hormonally active form of vitamin D	Hypocalcemia/abnormally low calcium blood levels, is in hypoparathyroidism. Hypercalcemia/abnormally high calcium blood levels- primary hyperparathyroidism	[1,3,7,10]
**Magnesium (Mg^2+^)**	ATP metabolism, muscle contraction and relaxation, proper neurological functioning, enzymatic reactions, nucleic acid synthesis, cell membrane ion transport, cell proliferation, calcium homeostasis, and neurotransmitter release. Mg^2+^ is involved in the secretion and activity of parathyroid hormone (PTH)	Its plasma concentrations depend on gastrointestinal absorption, renal excretion, and bone exchange	Hypomagnesemia-decreases in magnesium levels in the serum, may lead to gastrointestinal disorders	[1,3,6,10,11]

**Table 2 biology-11-01006-t002:** The reference values of selected electrolytes in the blood serum of animals.

**Animal**	**Electrolytes**					**Reference**
	**Cl^−^ [mg/dL]**	**Mg^2+^ [mg/dL]**	**K^+^ [mg/dL]**	**Na^+^ [mg/dL]**	**Ca^2+^ [mg/dL]**	[12]
**Cattle**	330–380	1.9–3.0	14.9–20.0	310–360	9.0–12.1	
**Horses**	320–380	1.7–2.8	13.5–18.5	320–360	10.7–13.4	
**Goats**	347–393	1.8–4.0	9.8–16.0	323–361	8.8–12.0	
**Sheeps**	345–400	2.0–3.0	16.0–20.0	340–370	10.0–13.0	
**Pigs**	340–390	2.3–3.5	17.0–22.0	320–360	8.0–16.0	

**Table 3 biology-11-01006-t003:** Daily demand for electrolytes for selected animal species.

Animal		Electrolyte					Literatures
		Sodium (Na^+^)	Potassium (K^+^)	Chloride (Cl^−^)	Calcium (Ca^2+^)	Magnesium (Mg^2+^)	
Cattle	Cows	0.11–0.20% (14–59 g/d)	1% (heifers 0.6%) 72–285 g/d	0.10–0.16%	45–210 g/d	16–50 g/d	[13]
	Bulls	0.11–0.20%	1%	0.10–0.16%	30–50 g	50 g	[13]
	Calves	0.15%	0.65%	0.20%	0.70%	0.10%	[14]
Pigs	Sow	1–1.2 g/d	0.2%	12–15 g/d	18–32 g/d	0.5–0.65 g/kg	[15,16,17]
	Porkers	1.3–1.7 g/d	0.2%	2.2–4.0 g/d	8–12 g/d	0.2–0.3 g/d	[15,16,17]
	Piglets	0.9%	0.2%	0.5–1.5 g/d	0.49% (3.3–5.9 g/d)	0.04%	[16,17,18]
Horse	Mare	27–62 g/d	15 g/d	67–123 g/d	26–28 g/d	6–14 g/d	[19]
	Stallion	27–62 g/d	15 g/d	67–123 g/d	26–28 g/d	6–14 g/d	[19]
	Foal	5–9 g/d	17–26 g/d	16–36 g/d	27–34 g/d	4–9 g/d	[19]
Poultry	Hen	0.14–0.17 g/d	0.2 g/d	0.3–0.45	2.8–4.7%	0.04%	[15]

Description: g/d—gram per day.

**Table 4 biology-11-01006-t004:** The normal plasma osmolality parameters for selected animals.

Animal	Range of Normal Plasma Osmolality	Reference
Cattle	270–310 mOsm/kg water	[24]
Horses	280–310 mOsm/kg water	[25]
Dogs	290–310 mOsm/kg water	[26]
Cats	290–330 mOsm/kg water	[27]
Adult human	275–295 mOsm/kg water	[8]

**Table 5 biology-11-01006-t005:** The role of selected electrolytes in the male and female reproductive systems.

Electrolyte	Male	Female	References
**Na^+^**	acrosomal reaction, sperm quality, sperm capacitation, motility	associated with the viability of the follicle and its activity in the synthesis of estrogens, participation in contraction, and relaxation of the uterus	[37,39,40,41,43,48,56,70,71,72]
**K^+^**	acrosomal reaction, sperm quality, motility	participation in contraction and relaxation of the uterus	
**Ca^2+^**	acrosomal reaction, sperm quality, sperm capacitation, motility, spermatogenesis, morphology of sperm	important role in the production of developing follicle hormones, regulation of the secretion of hormones necessary for ovaries and ovulation, estrogen synthesis, participation in contraction and relaxation of the uterus	
**Mg^2+^**	Sperm quality, sperm capacitation, motility, spermatogenesis, morphology of sperm	the function of nervous and muscle tissue, influence on lactation and growth of young animals, structure and development of bone tissue	
**Cl^−^**	Sperm motility, capacitation, acrosomal reaction, volume regulation	related to progesterone concentrations in mares and women; correlation with follicle diameter and estrogen concentration in buffaloes	

## Data Availability

Not applicable.

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
