# Peer review of "Reproductive Consequences of Electrolyte Disturbances in Domestic Animals"

_biology, 2022, doi:10.3390/biology11071006_

Round 1

Reviewer 1 Report

This is a bibliographic review paper. Perhaps the subject it covers is  excessively extensive and broad, which entails great difficulty.

The title is correct, and the text is generally well written.

These are comments about animals in general, and perhaps it could be indicated that he is more involved in domestic animals, in terms of work content.

The diagrams and tables are very illustrative and are of great help. Because sometimes the writing of the paragraphs is not easy, and at this point it is suggested to the authors that in the review they make an easier and less cumbersome writing. With not so long paragraphs.

It is suggested that if the order is to talk about the male first and then about the female, as in section 1, this order is followed in section 4, and this way it will be more coherent.

The bibliography is very extensive and up-to-date.

Author Response

Dear Reviewer,

As the authors of the manuscript, we would like to thank You for all general and detailed suggestions and guidelines as well as for pointing out the imperfections of the text.
Of course, we fully respected and applied all the detailed suggestions and additions sent by the all Reviewers to improve the text. Any corrections in the manuscript text are appropriately marked and apply to all suggestions, including those received from other reviewers.

Our responses to your comments are provided below:

Q1. This is a bibliographic review paper. Perhaps the subject it covers is  excessively extensive and broad, which entails great difficulty.The title is correct, and the text is generally well written. These are comments about animals in general, and perhaps it could be indicated that he is more involved in domestic animals, in terms of work content.The diagrams and tables are very illustrative and are of great help. Because sometimes the writing of the paragraphs is not easy, and at this point it is suggested to the authors that in the review they make an easier and less cumbersome writing. With not so long paragraphs.

Answer: Thank you for this valuable comment. Taking into account the opinion of both reviewers, we have made extensive changes to the text, fragments that are more difficult to understand we presented in tabular form, and the mechanisms important for a better understanding of the topic have been appropriately developed. As you suggested, we have modified the title to emphasize that this manuscript is about domestic animals.

Q2. It is suggested that if the order is to talk about the male first and then about the female, as in section 1, this order is followed in section 4, and this way it will be more coherent.

Answer: The text has been reorganized according to your suggestion.

Q3.The bibliography is very extensive and up-to-date.

Answer: Thank you.

Yours faithfully

Authors

Reviewer 2 Report

Electrolyte balance plays a key role in ensuring physiological function of organism. In this review, Galeska et al. introduced the functions, regulation and disorders of several major electrolytes in the body. The consequences of their disturbances in the reproduction is also discussed. The review paper should be written with suitable literature, detailed discussion, sufficient data/results to support the interpretation, and persuasive language style. However, the content and the title of the manuscript are less specialized and more generalized. Sufficient data and authors’ interpretation should be included in a good review paper. The reproduction system needs cooperation in many ways, most of the manuscript is related to hormone or other even non-reproduction work with less mechanism work. It is recommended to either rephrase the title or reorganize the main text.

Additional minor errors:

Line 42, potassium (K+)

The language is also needed to be edited meticulously and correctly to avoid unusual using or redundancy such as but not limited to line 26, 43-47, 80.

Author Response

Dear Reviewer,

As the authors of the manuscript, we would like to thank You for all general and detailed suggestions and guidelines as well as for pointing out the imperfections of the text.
Of course, we fully respected and applied all the detailed suggestions and additions sent by the all Reviewers to improve the text. Any corrections in the manuscript text are appropriately marked and apply to all suggestions, including those received from other reviewers.

Our responses to your comments are provided below:

Q1. Electrolyte balance plays a key role in ensuring physiological function of organism. In this review, Galeska et al. introduced the functions, regulation and disorders of several major electrolytes in the body. The consequences of their disturbances in the reproduction is also discussed.

The review paper should be written with:

  • suitable literature,
  • detailed discussion,
  • sufficient data/results to support the interpretation

Answer:  Taking into account the opinion of both reviewers and your very valuable comments, we have reorganized the text. The mechanisms important for a better understanding of the topic have been explained and adequately exhaustively described, as have been indicated in your opinion. For better clarity of the presented data, we have added an additional tables. We also supplemented the cited studies with specific and detailed data. Below you will find references to paragraphs that have changed significantly in terms of the content presented.

  • line 246-251: added data about the changes in the electrolyte profile of mares during ovulation
  • line 257-270: added data about the changes in the electrolyte profile of women during the menstrual cycle
  • line 279-315: added data about the effect of calcium on follicle growth in birds with the description of the mechanisms of calcium action during the hierarchical follicles selection
  • line 323-348: the description of the mechanisms responsible for the activation of oocytes in the presence of electrolytes and the description of effect calcium inhibitors on mice oocyte maturation in vitro
  • line 353-361: the description of research carried out on camel’s ovary – compared the content of sodium and potassium in small and large follicles, and the influence of those electrolytes on the development of the follicles
  • line 389-399: authors cited the research on roosters in which the content of calcium, magnesium, phosphorus, iron, and copper where analyzed in terms of the sperm viability and motility
  • line 401-415 added the data about relationship between buck’s semen quality (semen volume, sperm motility, percentage content of sperm) and the content of sodium, potassium, chloride, calcium, phosphorus in the semen plasma.
  • line 420-444: added the description of the effect of Na, K-ATPase α4 knockout in mice semen in terms of infertile male and the effect of imbalance of sodium levels in semen
  • line 446-482: the authors extended their manuscript to describe the action of ions in sperm capacitation
  • line 485-513: extended description by comparing the effect of sperm plasma electrolytes on sperm quality in fertile and infertile men with varicocele and after varicocele surgery
  • line 516-541: added a description about the effect of supplementation of bucks with copper and zinc on quality of semen with the data from the cited article
  • line 645-648: extending of the description of the effect of sodium on mortality in cats with specific values
  • line 675-683: the cited studies of pig electrolyte supplementation during pregnancy were extended with specific numerical values of the effect of supplementation on litter size at birth, average birth weight and average weaning weight, as well as on the born alive index
  • line 717-724: extended of the description of the research on bovine follicular fluid and the content of electrolytes
  • line 731-742: extended of the description of influence of calcium during conception and the sperm specific phospholipase C zeta on calcium oscillations, mechanism of calcium action during the oocyte activation and the description of the effect of calcium stimulants on artificial fertilization to induce oocyte activation

  • and persuasive language style.

Answer: We have tried to improve the language in this text. If, despite our efforts, the reviewer still notices errors, we will send the manuscript to the MDPI language edition.

Q2. However, the content and the title of the manuscript are less specialized and more generalized. Sufficient data and authors’ interpretation should be included in a good review paper. The reproduction system needs cooperation in many ways, most of the manuscript is related to hormone or other even non-reproduction work with less mechanism work. It is recommended to either rephrase the title or reorganize the main text.

Answer: In line with your suggestion and taking into account the opinion of the second reviewer, we have reorganized the text and modified the title of the manuscript. As we mentioned in previous comments, we have described the mechanisms and supplemented the cited studies with specific data.

Q3. Line 42, potassium (K+)

Answer: It has been corrected.

Q4. The language is also needed to be edited meticulously and correctly to avoid unusual using or redundancy such as but not limited to line 26, 43-47, 80.

Answer: We have tried to improve the language in this text. If, despite our efforts, the reviewer still notices errors, we will send the manuscript to the MDPI language edition.

Yours faithfully

Authors

Round 2

Reviewer 2 Report

The manuscript was revised well, there is no more comment.